# THE BACTERIAL KINGDOM IN HYPERBOLIC SPACE

## ABSTRACT

The bacterial kingdom remains largely unexplored, with new strains continuously being discovered. The exponential growing size of bacterial databases poses the need for succinct yet informative representations of these vast microbial collections, allowing for fast and efficient genome classification and comparison. To address this, we propose HyperBiome, a metric learning framework that takes advantage of the geometry of the Poincarè ball to reconstruct the bacterial taxonomy and compute a latent space where distances reflect biological similarities between genomes. By incorporating the taxonomic hierarchy in hyperbolic space, we learn representative proxies at both the species and genus level, which guide the embedding of each bacterial assembly. Finally, using the species-level proxies, we build a compact index that enables rapid classification of new assemblies while avoiding exhaustive query-vs-all scans of the database. Experiments on AllThe-Bacteria, the largest bacterial database, demonstrate that HyperBiome effectively captures biological relationships. Moreover, we show that our proxy-based index achieves high accuracy, substantially reduces computational querying costs, and generalizes successfully to previously unseen species, supporting continuous updates without retraining the metric model.

## 1 INTRODUCTION

A comprehensive understanding of the bacterial domain is fundamental in life sciences, with important implications across many application fields including public health monitoring (Foxman et al., 2025) and medicine (Hou et al., 2022). Modern high-throughput sequencing technologies allow the extraction of thousands of bacterial genomes from environmental and clinical samples, with assembly processes achieving high levels of accuracy in reduced time frames (Ekim et al., 2021). This rapid and massive data production has led to a huge expansion of bacterial genomic databases (Bradley et al., 2019), making them more representative of microbial diversity but computationally demanding for thorough exploration. Consequently, substantial research efforts have focused on developing compressed bacterial collections (Marchet et al., 2021; Zhu et al., 2015) that preserve representative power while enabling easier access, efficient searching, and scalable comparative analyses. However, generating such succinct representations poses substantial challenges due to the immense volume of data and the extraordinary evolutionary diversity across bacteria. Moreover, current databases capture only a small fraction of the bacterial diversity present in nature, and their size is expected to grow exponentially in the coming years, further intensifying these challenges. For this reason, by leveraging the standard string representation of DNA, combinatorial techniques (originally developed for text compression) are increasingly applied to generate succinct data structures for genomic datasets (Mäkinen et al., 2023). Additionally, deep learning techniques have shown great promise in generating precise representations of genomes across different applications (Routhier & Mozziconacci, 2022). Moreover, the inherent hierarchical nature of genomic data has proven to be a powerful asset for maximizing the compression of traditional indices (Břinda et al., 2025). Recently, this property has also been explored to integrate DNA sequences within geometric models (Tian et al., 2023), such as hyperbolic spaces, which are particularly well-suited to capturing intrinsic hierarchical structures.

**The HyperBiome Framework.** In this paper, we propose HyperBiome[1], a framework that maps the bacterial domain to a geometric space that reflects the intrinsic biological organization. Bacterial

---

[1]The code is available on GitHub: `https://anonymous.4open.science/r/Hyperbiome-431F`

taxonomy is inherently tree-structured, and organisms are organized into nested categories such as *phyla*, *classes*, *genera*, and *species*. Higher levels of taxonomy (e.g., phylum, class) capture broad evolutionary divergence, while the genus and species levels describe finer-grained relationships. A genus groups species that share a close evolutionary lineage, while a species is the most specific unit of classification, representing a population of strains with very high genetic similarity. Within HyperBiome, bacterial representations are projected into a hyperbolic space, using a loss function that guides the metric learning process to preserve the taxonomic structure. Thus, HyperBiome exploits the tree-likeness property of hyperbolic spaces, which naturally support the encoding of hierarchical relationships in continuous space (Nickel & Kiela, 2018; Sonthalia & Gilbert, 2020).

## 2 PRELIMINARIES

### 2.1 SEARCHING THROUGH GENOMIC DATABASES

Bacterial databases maintain collections of strings drawn from the alphabet $\Sigma = \{A, C, G, T\}$. Each sequence represents the linear arrangement of nucleotide bases in the corresponding bacterial genome. Searching for biologically related sequences to a given query is a fundamental task in many downstream applications, including phylogenetic and evolutionary analyses, as well as species classification. For many years, *BLAST* (Madden, 2013) has been considered the standard tool for this type of analysis. However, it relies on pairwise sequence alignments, which are computationally expensive. Despite recent speedups (Shen et al., 2025), alignment-based approaches remain a bottleneck when exploring modern large-scale genomic databases. These limitations have motivated the development of alignment-free methods, which exploit the statistical properties of $k$-mers, defined as all contiguous sequences of length $k$ extracted from a genome. By comparing $k$-mer distributions across genomes, one can efficiently estimate biological similarity and evolutionary relationships while avoiding full sequence alignments (Zielezinski et al., 2017). This approach also enables representing the sequence collection using compressed indices based on $k$-mer, using data structures for text compression and optimized for pattern matching queries (Bingmann et al., 2019; Schiff et al., 2024). A crucial factor in this process is the choice of $k$, which depends on the specific application. When comparing assembled bacterial genomes, larger $k$ values generally improve resolution as they capture more distinctive signatures. However, the cardinality of the $k$-mer space grows exponentially with $k$, leading to significant computational challenges that can limit the benefits of alignment-free approaches.

### 2.2 EFFICIENT BACTERIAL GENOMES INDEXING VIA METRIC LEARNING

Learned indices and deep learning methods have proven to be powerful solutions for large-scale data retrieval, offering enhanced search performance and compact representations(Ferragina et al., 2020; Chen et al., 2022). Since nucleotide sequences are naturally represented as strings, large language models (LLMs) offer a natural framework for learning from genomic data. Indeed, several LLM architectures have been specifically developed to model the language of DNA, demonstrating promising results in uncovering both structural and functional properties of genomes (Avsec et al., 2025; Nguyen et al., 2023; Zhou et al., 2023). Moreover, advances in state-space modeling have enabled these models to handle increasingly long sequences and capture complex dependencies (Gu & Dao, 2023; Schiff et al., 2024). However, despite these improvements, training and inference remain computationally prohibitive for complete genome assemblies, which often comprise billions of base pairs. Consequently, LLMs remain too resource-intensive to be directly applied for building lightweight indices capable of efficiently querying millions of bacterial complete genomes. To address this challenge, PanSpace (Cartes et al., 2025) proposes a novel vector index in which distances between elements reflect biological similarity. In particular, PanSpace is a framework based on a convolutional neural network that encodes bacterial assemblies into dense embeddings using an image-based representation of the distribution of $k$-mers derived from the frequency chaos game representation (FCGR) (Löchel & Heider, 2021). These embeddings are learned using a triplet metric learning loss function and organized within a FAISS index (Douze et al., 2024), enabling fast nearest-neighbor searches. By projecting genomes into an embedding space, PanSpace enables an efficient exploration of large bacterial collections. It outperforms the state-of-the-art tools (Zhao et al., 2024) in memory usage and query performance, with only minimal degradation in species-level classification accuracy.

## 2.3 CAPTURING HIERARCHIES IN HYPERBOLIC SPACES

Recent studies suggest that genomic data exhibit intrinsic hierarchical structures shaped by evolution (Corso et al., 2021; Khan et al., 2025), making hyperbolic space a natural geometry for learning models (Ganea et al., 2018) that capture genome organization efficiently. In particular, bacterial genomes display a tree-like taxonomic organization, which includes multiple hierarchical levels leading to *genera* and then branching further into *species*. A hyperbolic space is a Riemannian manifold with constant negative curvature. More generally, a Riemannian manifold $(\mathcal{M}, g)$ is a differentiable manifold $\mathcal{M}$ on which a Riemannian metric $g$ is defined. An $n$-dimensional manifold $\mathcal{M}$ is a topological space that is locally approximated by $\mathbb{R}^n$. Thus, for each point $x \in \mathcal{M}$, the tangent space $T_x\mathcal{M}$ is defined as the vector space representing the first-order linear approximation of $\mathcal{M}$ near x. The curvature of a Riemannian manifold quantifies how much the geometry deviates from that of flat Euclidean space. The Riemannian metric $g$ defines the infinitesimal geometry of tangent spaces and a globally consistent notion of the distance across the manifold. Operations on hyperbolic spaces are based on geodesics and gyrovector formalism, extending beyond the familiar notions of Euclidean vector algebra. Intuitively, geodesics describe the shortest paths between two points on the manifold and gyrovectors represent how to move from one point to another along a geodesic. In the negative curvature case, the geodesics diverge exponentially, giving rise to a geometry that is *saddle-shaped* at every point. In Riemannian geometry, it is often useful to project points from the manifold to its tangent space, and viceversa. Indeed, the tangent space $T_x\mathcal{M}$ is a Euclidean vector space of the same dimension of $\mathcal{M}$, allowing to perform computations on a vector space locally, even if the global geometry of the manifold is curved. The tangent space at a point $x \in \mathcal{M}$ and the manifold $\mathcal{M}$ itself are related via the exponential and logarithmic maps, which provide local bijections between them:

$$\exp_x : T_x\mathcal{M} \rightarrow \mathcal{M}, \quad \log_x : \mathcal{M} \rightarrow T_x\mathcal{M}. \tag{1}$$

Here, $\exp_x$ maps a tangent vector $v \in T_x\mathcal{M}$ to a point $x \in \mathcal{M}$ along the corresponding geodesic, while $\log_x$ is its local inverse, mapping points on the manifold back to the tangent space at $x$.

**The Poincarè ball.** Several models have been developed to represent hyperbolic spaces. Among these, the Poincarè ball model has demonstrated success in improving metric learning performance on data with non-Euclidean structure, while maintaining simple and closed-form geometric formulations (Yan et al., 2021). The Poincarè ball $\mathcal{B}_c^D$ with the curvature parameter $c$ provides a realization of $D$-dimensional hyperbolic geometry, where the entire space is represented inside the open unit ball of $\mathbb{R}^n$ with radius $r = \frac{1}{\sqrt{c}}$:

$$\mathcal{B}_c^D = \{x \in \mathbb{R}^D : c \|x\|^2 < 1, c \geq 0\} \tag{2}$$

and the Riemannian metric is:

$$\mathfrak{g}_c^x = (\lambda_c^x)^2 \mathbf{I}_n \tag{3}$$

where the conformal factor $\lambda_c^x = 2\left(1 - c\|\mathbf{x}\|^2\right)^{-1}$ is the scaling factor for the Euclidean metric tensor. The geodesic distance between $\mathbf{x}, \mathbf{y} \in \mathcal{B}_c^n$ is defined as:

$$d_c(\mathbf{x}, \mathbf{y}) = \frac{2}{c} \tanh^{-1}\left(c \left\| \ominus_c \mathbf{x} \oplus_c \mathbf{y} \right\|\right) = \left\| \log_{\mathbf{x}}^c(\mathbf{y}) \right\|_{\mathbf{x}}^c \tag{4}$$

with $\oplus$ being the Möbius addition:

$$\mathbf{x} \oplus_c \mathbf{y} = \frac{\left(1 + 2c\langle\mathbf{x}, \mathbf{y}\rangle + c\|\mathbf{y}\|^2\right)\mathbf{x} + \left(1 - c\|\mathbf{x}\|^2\right)\mathbf{y}}{1 + 2c\langle\mathbf{x}, \mathbf{y}\rangle + c^2\|\mathbf{x}\|^2\|\mathbf{y}\|^2}, \tag{5}$$

which reverse calculation is defined by

$$\mathbf{x} \ominus_c \mathbf{y} = \mathbf{x} \oplus_c (-\mathbf{y}). \tag{6}$$

Being a geodesically complete manifold, the Poincarè ball is characterized by closed-form expressions for the exponential and logarithmic maps. The exponential map $\exp_{\mathbf{x}}^c : \mathcal{T}_{\mathbf{x}}\mathcal{B}_c^D \to \mathcal{B}_c^D$ is defined as follows:

$$\exp_{\mathbf{x}}^c(\mathbf{v}) = \mathbf{x} \oplus_c \frac{1}{c} \tanh\left(c\,\lambda_{\mathbf{x}}^c\,\|\mathbf{v}\|\right)[\mathbf{v}], \quad \forall \mathbf{x} \in \mathbb{B}_c^D,\ \mathbf{v} \in \mathcal{T}_{\mathbf{x}}\mathcal{B}_c^D. \tag{7}$$

The logarithmic map $\log_{\mathbf{x}}^c = (\exp_{\mathbf{x}}^c)^{-1} : \mathcal{B}_c^D \to \mathcal{T}_{\mathbf{x}}\mathcal{B}_c^D$ provides the inverse operation:

$$\log_{\mathbf{x}}^c(\mathbf{y}) = \frac{2}{c}\,\lambda_{\mathbf{x}}^c\,\tanh^{-1}\left(c\,\|\ominus_c\,\mathbf{x}\oplus_c\mathbf{y}\|\right)[\ominus_c\mathbf{x}\oplus_c\mathbf{y}], \quad \forall \mathbf{x},\mathbf{y} \in \mathcal{B}_c^D. \tag{8}$$

## 3 HYPERBIOME: A HYPERBOLIC FRAMEWORK FOR EMBEDDING BACTERIAL KINGDOM

### 3.1 GENOME TENSORIAL SKETCHING

HyperBiome learns a representation of bacterial genomes in a metric space starting from a compact encoding of their distribution of $k$-mers (see Section 2.1). HyperBiome employs a tensorial sketching approach to generate vectorial summaries of bacterial genomes. Genome tensorial sketching (Joudaki et al., 2020) maps nucleotide sequences into $D$-dimensional vectors preserving compositional information of a representative sample of $k$-mers. This approach prevents combinatorial growth in representation size, enabling generalization to longer and more specific $k$-mers. In particular, HyperBiome employs the hyperdimensional sketching strategy proposed by Xu et al. (2024). Given an input genomic assembly $a$, the corresponding set $\mathcal{K}$ of unique $k$-mers is extracted, and each $k \in \mathcal{K}$ is mapped to a numerical value in the range $[0, M]$ using a hash function $h$. Then, a subset $\mathcal{I}_\beta$ of $k$-mers is selected using the *FracMinHash* technique (Pierce et al., 2019) as follows:

$$\mathcal{I}_\beta = \{h(k) \mid k \in \mathcal{K},\ h(k) \le M/\beta\} \tag{9}$$

where $\beta > 1$ is a scale factor controlling the fraction of $k$-mers retained in the sketch. After the sampling procedure, each $i \in \mathcal{I}_\beta$ is turned into a vector $b \in \{0,1\}^D$. The binary vector is constructed in $D/64$ blocks. Each block corresponds to a 64-bit value generated by a random number generator seeded with $i$. This ensures that each element is associated with a unique, deterministic and pseudo-random binary vector. We denote the collection of binary vectors obtained from $\mathcal{I}_\beta$ as $\mathcal{V}$. Finally, all elements of $\mathcal{V}$ are mapped from $\{0,1\}^D$ to $\{-1,1\}^D$ and, then, aggregated into a non-binary $D$-dimensional sketch summarizing the input sequence $a$. This is done by performing a point-wise vectorial summation:

$$sketch(a) = \sum_{v \in \mathcal{V}} \left(2 \cdot v - 1\right)$$

### 3.2 PROJECTING BACTERIA TO POINCARÈ BALL

HyperBiome implements a metric learning architecture that projects bacterial assembly sketches into the Poincarè ball model, effectively capturing hierarchical structures and the inherent organization of bacterial taxonomy. To achieve this, the framework leverages the exponential map (see Equation 7) to establish a bijection between the Euclidean tangent space and the hyperbolic manifold. This allows optimizing network parameters in the tangent space while preserving the representational advantages of hyperbolicity in the Poincarè ball model (Ermolov et al., 2022).

**Embedding Model.** To obtain a lower-dimensional embedding, each $D$-dimensional tensorial sketch (see Section 3.1) is split into non-overlapping patches. Let $p$ denote the length of each patch, and let $N = D/p$ be the total number of patches. The sketch is reshaped into a patch matrix:

$$\boldsymbol{A} \in \mathbb{R}^{N \times p}$$

Each patch is then linearly projected into a $d$-dimensional space $(d < D)$ using a shared linear projection layer:

$$Z_0 = \text{Linear}(\boldsymbol{A}) \in \mathbb{R}^{N \times d}$$

A learnable class token $c \in \mathbb{R}^{1 \times d}$ is prepended to the sequence, and a learnable positional embedding $E_{\text{pos}} \in \mathbb{R}^{(N+1) \times d}$ is added to retain positional information:

$$Z_0' = [c; Z_0] + E_{\text{pos}} \in \mathbb{R}^{(N+1) \times d}$$

The sequence is processed by a stack of four Transformer encoder layers, each consisting of four-head self-attention and a feed-forward network with hidden dimension 256. The final embedding is extracted from the output corresponding to the class token:

$$z = Z_L[0] \in \mathbb{R}^d,$$

where $Z_L$ denotes the output of the last Transformer layer. Finally, $z$ is projected onto the Poincarè ball $\mathcal{B}_c^d$. HyperBiome assumes that $z$ lies in the tangent space $T_0 \mathbb{B}_c^d$ and thus, applies the exponential map at the origin. Thus, Formula 7 becomes:

$$\exp_0^c(\mathbf{v}) = \tanh\left(\sqrt{c} \, \|\mathbf{v}\|\right) \frac{\mathbf{v}}{\sqrt{c} \, \|\mathbf{v}\|}.$$

Before applying the exponential map HyperBiome enforces a clipping radius $\tau$ to ensure numerical stability, as suggester by Guo et al. (2022). The parameter $\tau$ defines a maximum Euclidean radius of the tangent vector, preventing it from being mapped too close to the boundary of the Poincarè ball. Near the boundary, distances grow rapidly (see Formula 4) and floating-point errors may cause points to numerically escape the manifold. Clipping keeps embeddings within a safe region, preserving both stability and the expressive geometry of hyperbolic space.

**Hierarchical Multi-Proxy Loss.** Within HyperBiome, the projection onto the $\mathcal{B}_c^d$ results from the minimization of a reformulated version of the Proxy-Anchor (PA) loss (Kim et al., 2020). Unlike other proxy-based metric learning losses, the PA loss offers significant computational advantages while preserving the relational structure among data points. Basically, its standard formulation assigns each class a representative proxy and encourages embeddings to remain close to their corresponding positive proxies, while simultaneously being pushed away from negative proxies, with magnitudes that reflect inter-sample relationships. HyperBiome extends the PA loss by introducing a two-level hierarchical proxy system designed to capture genera–species taxonomic relationships (see Section 2.3). Let $P_s = \{p_s\}_{s=1}^{n_{\text{species}}}$ denote the set of species-level proxies. We define the species-level loss as:

$$
\begin{aligned}
\mathcal{L}_{\text{species}} = \frac{1}{|P_s^+|} \sum_{p \in P_s^+} \sum_{x \in X_p^+} \log(1 + e^{d_c(x,p)}) \\
+ \frac{1}{P_s} \sum_{p \in P_s} \sum_{x \in X_p^-} log(1 + e^{-d_c(x,p)})
\end{aligned}
\tag{10}
$$

where $d_c(x,p)$ is the distance between a sample of the batch $x$ and a species-level proxy $p$ on $\mathcal{B}_c^d$ (see Formula 4), $P_s^+$ is the set of positive species-level proxies in the batch, $X_p^+$ and $X_p^-$ are the set of positive and negative samples of the proxy $p$ in the batch, respectively. Then, let $P_g = \{p_g\}_{g=1}^{n_{\text{genera}}}$ denote the set of genus-level proxies and let $\pi : \{1, \ldots, n_{\text{species}}\} \to \{1, \ldots, n_{\text{genera}}\}$ map each species index to its corresponding genus index. The genus-level loss is then defined as:

$$
\begin{aligned}
\mathcal{L}_{\text{genus}} = \frac{1}{|P_g^+|} \sum_{p \in P_g^+} \sum_{x \in P_s \cap X_p^+} \log(1 + e^{d_c(x,p)}) \\
+ \frac{1}{|P_g|} \sum_{p \in P_g} \sum_{x \in P_s \cap X_p^-} log(1 + e^{-d_c(x,p)})
\end{aligned}
\tag{11}
$$

where $P_g^+$ is the set of positive genus-level proxies in the batch, $P_s \cap X_p^+$ and $P_s \cap X_p^-$ are the set of positive and negative species-level proxies of the genus-level proxy p, respectively. Finally, the total hierarchical multi-proxy (HMP) loss combines the two levels:

$$\mathcal{L}_{\text{HMP}} = \mathcal{L}_{\text{species}} + \mathcal{L}_{\text{genus}}. \tag{12}$$

Thus, minimizing the HMP loss encourages embeddings of bacterial species to remain close to the proxies of their corresponding species, while species-level proxies are in turn constrained to remain close to the proxies of their parent genera.

### 3.3 PROXY-BASED INDEX.

Analogously to PanSpace, the HyperBiome framework encodes bacterial representations into a compact $d$-dimensional space, facilitating efficient compressed indexing and rapid database exploration. However, HyperBiome offers a hierarchical view of the bacterial database, yielding a more informative and structured representation of the domain organization. Each learned genus-level proxy summarizes the corresponding set of species. Finally, bacterial embeddings are further grouped into species-level clusters, each represented by a learned species-level proxy. In short, we can perform $m$-nearest neighbor retrieval operations, this time considering hyperbolic distances on $\mathcal{B}_c^d$, which better exploit the hierarchical organization within species and across species and genera. Moreover, by learning synthetic proxies for each species, HyperBiome enables the generation of a proxy-based index that can be used to classify the species of new assemblies without requiring comparison against all the stored genomes. Instead, classification can be achieved by computing only the hyperbolic distance between the new assembly and the learned species-level proxies, assigning the species corresponding to the nearest proxy. This index can also be updated efficiently with the addition of new species, without involving costly retraining procedures to include new classes. A new proxy can be generated once the corresponding new assemblies have been embedded into $B_c^d$ according to the learned metric model, and their hyperbolic centroid can be used as the species-level proxy. Specifically, given a set of embeddings $\{x_i\}_{i=1}^n$ in hyperbolic space $\mathbb{H}^d$, the Fréchet mean $\mu$ is defined as the minimizer of the sum of squared hyperbolic distances:

$$\mu = \arg \min_{y \in \mathbb{H}^d} \sum_{i=1}^{n} d_{\mathbb{H}}^2(y, x_i), \tag{13}$$

where $d_{\mathbb{H}}$ denotes the hyperbolic distance metric. This mean can be computed iteratively using gradient descent or other optimization techniques adapted to hyperbolic geometry.

## 4 EXPERIMENTAL ANALYSIS

In this section, we evaluate the performance of HyperBiome in embedding the largest publicly available collection of bacterial sequences. We systematically assess the representational capacity of the resulting hyperbolic latent space and its effectiveness in reconstructing bacterial domain organization. Additionally, we assess the efficacy of the proxy-based indexing strategy for strain classification, focusing on both output quality and querying efficiency.

**Dataset.** We conducted our experiments on the most recent version of the *AllTheBacteria* database (Hunt et al., 2024), the most comprehensive collection of bacterial sequences, containing several terabytes of genomic data. This database provides a broad up-to-date representation of bacterial diversity, but it is highly unbalanced, with a predominance of well-studied species, particularly those of medical and healthcare relevance. Indeed, although the database contains $11,824$ species, approximately $75\%$ of the assemblies are concentrated in just 10 species. To ensure data quality and meaningful evaluation, we filtered out unclassified and low-quality genomes, retaining only 179 species across 62 genera [2]. In our setting, the most represented species contains $426,587$ assemblies and the least represented species containing contains 200 assemblies. Further, we partitioned the dataset into two subsets. The *seen set* comprises $1,798,003$ genomes from the $154$ most-represented species in the dataset, whereas the *unseen set* includes $5,529$ strains distributed across 25 species. Note that all the genera are represented in both subsets, but by different species. To evaluate the HyperBiome framework, both the seen and unseen sets were partitioned into a gallery and a query set using a standard 80–20 split. The *seen gallery* was used to train the HyperBiome model, while the *seen queries* were used to evaluate performance on species observed during training. In contrast, the *unseen gallery* was used to augment the embedding space and update the proxy-based index

---

[2]For reproducibility, the code used to sample the dataset employed in our experiments is available on anonymous Github repository

with species not encountered during the training. Finally, the *unseen queries* were used to evaluate the effectiveness of the updating strategy (see Section 3.3) for the proxy-based index in constructing proxies for previously unseen species.

**Training Setting.** We implemented HyperBiome in PyTorch and trained it for 100 epochs with a batch size of 32 and a learning rate of $10^{-4}$, using the Adam optimizer. Training incorporated a *ReduceLROnPlateau* scheduler (mode set to *min*, factor 0.5, patience 3, threshold $10^{-4}$, cooldown 0, and minimum learning rate $10^{-7}$). Early stopping was applied with a patience of 10 epochs. For PanSpace, we used the public TensorFlow implementation and trained it with their default settings available at `https://github.com/pg-space/panspace`. Concerning the choice of $k$, longer $k$-mers generally yield more precise genomic signatures, as they capture richer sequence-specific information. Authors of PANSPACE suggested $k = 7$. However, considering larger values is impractical, as the method relies on images of size $2^k \times 2^k$. This results in a combinatorial explosion of dimensions that makes both training and indexing computationally intractable. In contrast, HyperBiome leverages tensorial sketching, which enables us to scale up to $k = 21$. This allows for a more expressive representation of $k$-mer distributions, effectively tripling the contextual resolution.

## 4.1 EXPERIMENT 1: EVALUATING THE IMPACT OF CURVATURE AND CLIPPING RADIUS

**Goal and Experimental Design.** In this experiment, we investigated how the curvature parameter $c$ and the clipping radius $\tau$ affect the performance of HyperBiome. Specifically, we evaluated the accuracy of the proxy-based index in classifying both seen and unseen queries at the species-level under varying parameter configurations. The primary goal was to identify the optimal combination that balances output quality and generalization performances. To this end, we conducted a controlled grid search over selected values of $c$ and $\tau$, training HyperBiome with each configuration and constructing the corresponding proxy-based. The proxies for species in the unseen gallery were computed using the Frechet mean update (see Section 3.3). Performances were evaluated on both seen and unseen queries in terms of species-level retrieval accuracy, defined as the proportion of genomes correctly assigned to the proxies corresponding to their respective species. This systematic evaluation allowed us to assess the role of $c$ and $\tau$ in shaping the capacity of HyperBiome to summarize species into learned proxies and generalize to new species.

**Choice of $c$ and $\tau$.** Regarding the curvature parameter $c$, as it tends to zero, the radius of the Poincarè ball tends toward infinity, collapsing to a Euclidean space. Conversely, larger values of $c$ increase the hyperbolicity of the embedding space, thereby affecting the hierarchical structure that can be represented. On the other side, the parameter $\tau$ serves as a stabilization mechanism, preventing vectors from being mapped too far from the origin and avoiding numerical instabilities due to floating-point errors. In this study, the clipping radius was defined as a fraction of the maximum allowable Euclidean radius $r$ for each curvature, following three strategies. The aggressive strategy adopts a larger radius ($\frac{3}{4}r$), preserving hierarchical structure but potentially reducing stability. Then, the moderate strategy ($\frac{1}{2}r$) balances expressivity and stability. Finally, the conservative strategy ($\frac{1}{4}r$) imposes tighter constraints that enhance stability and regularization at the expense of representational capacity. To identify the optimal $(c, \tau)$ combination, the model was trained on the seen gallery while varying $c$ and testing each clipping strategy.

**Results.** Table 1 summarizes the species-level retrieval accuracy for different combinations of curvature parameter $c$ and clipping radius $\tau$. For seen queries, performance remains consistently high across all configurations, exceeding 99.6% in most cases, indicating that the model builds robust and significative species-level proxies data regardless of hyperparameter choice. Conversely, for the unseen queries, the impact of $c$ and $r$ is more pronounced. At the smallest curvature ($c = 0.01$), increasing the clipping radius significantly improves generalization, with accuracy rising from 0.8933 at $r = 2.5$ to a peak of 0.9530 at $r = 7.5$. This suggests that lower curvature with a larger radius allows the embedding space to better capture biological relationships and generalize to unseen species, making the learned metric more effective at representing new species by summarizing them with their own hyperbolic centroid. At higher curvatures ($c = 0.05$ and $c = 0.10$), although seen-query accuracy remains near optimal, unseen-query performance generally declines, particularly for smaller values of $\tau$. For $c = 0.05$, the best unseen performance (0.9277) occurs at the largest tested clipping radius ($\tau = 3.35$), while for $c = 0.10$, there are no improvements while increasing $\tau$. These

results show that smaller curvature combined with a more permissive clipping strategy offers the best trade-off between fitting the seen data and generalizing to unseen queries, with $c = 0.01, r = 7.5$ emerging as the optimal configuration in this experimental setting.

| c | r | Clipping Strategy | $\tau$ | Accuracy | |
|---|---|---|---|---|---|
| | | | | **Seen Gallery** | **Unseen Gallery** |
| 0.01 | 10 | conservative | 2.5 | 0.9969 | 0.8933 |
| | | moderate | 5.0 | 0.9979 | 0.9503 |
| | | aggressive | 7.5 | 0.9971 | **0.9530** |
| 0.05 | 4.47 | conservative | 1.12 | 0.9995 | 0.8553 |
| | | moderate | 2.24 | 0.9996 | 0.8807 |
| | | aggressive | 3.35 | 0.9983 | 0.9277 |
| 0.10 | 3.16 | conservative | 0.79 | 0.9992 | 0.8734 |
| | | moderate | 1.58 | 0.9996 | 0.8146 |
| | | aggressive | 2.37 | **0.9997** | 0.8698 |

Table 1: Species-level classification accuracy of the proxy-based index resulting from training the HyperBiome for different ($c$,$r$) combinations.

## 4.2 EXPERIMENT 2: COMPARISON WITH PANSPACE

**Goal and Experimental Design.** In this experiment, we compared PanSpace against the proxy-based index derived from the best-performing HyperBiome configuration $(c, r)$ identified in Experiment 1. Both methods are evaluated in terms of their accuracy in assigning the correct species to both seen and unseen queries. For seen queries, each method relies on its respective representation of the seen gallery dataset. For unseen queries, the proxy-based derived from HyperBiome index naturally accommodates novel species through a continuous update mechanism (see Section 3.3). In contrast, PanSpace does not provide a dedicated update strategy. We incorporated new bacterial genomes into the vector index provided by PanSpace using the embedding model trained on the seen gallery dataset. Species-level classification is performed differently in the two frameworks. HyperBiome assigns queries to the nearest proxy, whereas PanSpace identifies the $m$-nearest neighbors and infers the species from the most prevalent class among them. As reported by the authors of PanSpace, $m = 1$ yields the optimal classification performance, corresponding to assigning the query to the class of the single nearest neighbor. Additionally, while the original PanSpace implementation adopts an embedding dimensionality of $d = 256$, we retrain HyperBiome with a reduced dimensionality of $d = 128$, thereby achieving a more compact representation. For consistency, we also evaluate PanSpace under the same dimensionality setting.

| Method | Dimension | Accuracy | |
|---|---|---|---|
| | | **Seen Gallery** | **Unseen Gallery** |
| Hyperbiome | 128 | 0.9971 | **0.9530** |
| Panspace | 128 | **0.9984** | 0.8505 |
| Panspace | 256 | 0.9775 | 0.5634 |

Table 2: Comparison of species-level classification accuracy between HyperBiome and PanSpace

**Results.** The results in Table 2 highlight a clear trade-off between the two approaches. Hyper-Biome, through its proxy-based index, achieves high accuracy on both seen and unseen queries, with only a marginal drop when extending to new species. This property is particularly valuable because the index can be continuously updated with novel proxies without requiring expensive full or partial retraining. Additionally, the computational cost of classification is substantially reduced since queries are compared only against the number of species-level proxies, which keeps the index compact and the complexity of inference low. On the other hand, PanSpace achieves excellent accuracy on seen species, slightly outperforming HyperBiome in this setting. However, it fails to generalize effectively to unseen species, as performance drops significantly on unseen queries especially for larger values of $d$. Moreover, in scenarios where only species-level classification is required, PanSpace becomes less practical since it requires storing all bacterial embeddings in the index and performing exhaustive all-against-all comparisons, which is both computationally expensive and memory-demanding compared to maintaining a single proxy per species.

### 4.3 EXPERIMENT 3: RETRIEVAL SPACE ORGANIZATION

**Goals and Experimental Design.**     When querying bacterial databases, an important aspect beyond species-level classification is the quality of retrieval. A key property of any retrieval framework lies in how points belonging to the same class are organized within the embedding space. To investigate this property, we performed a large-scale retrieval space analysis by sampling $20,000$ genomes from the 20 most represented species in our gallery. For PanSpace, we generated two-dimensional embeddings of this subset using UMAP with Euclidean distance. For HyperBiome, we applied the same procedure but employed the hyperboloid metric, reflecting the underlying geometry. This setup enabled a direct graphical comparison of the two frameworks in terms of their ability to preserve species-level clustering and to maintain strains of the same species in close proximity.

**Results.**     Figure 1 and Figure 2 reveal substantial differences in how PanSpace and HyperBiome learn the embedding space. In PanSpace (Figure 1), although species are roughly separable, the clusters are often irregular and fragmented. In some cases, assemblies from the same species are split into multiple disconnected regions of the plane. This fragmentation weakens intra-species cohesion. In contrast, HyperBiome (Figure 2) produces well-formed, compact clusters, with all assemblies of the same species grouped together and clearly separated from other species. The clusters exhibit a radially organized arrangement, reflecting the geoemtry of Poincarè ball. This structure not only strengthens species-level cohesion but also provides clearer inter-species boundaries, making retrieval more reliable and biologically interpretable.

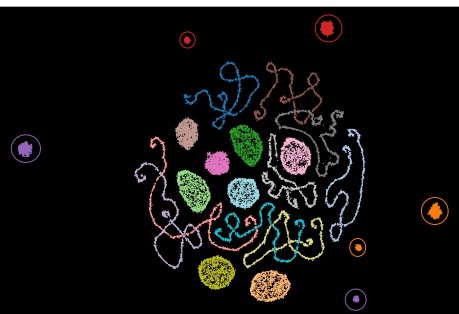

Figure 1: Two-dimensional representation of the embedding space generated by PanSpace using UMAP. Each color corresponds to a different bacterial species. Clusters circled with the same color belong to the same species but are partitioned across different regions of the plane.

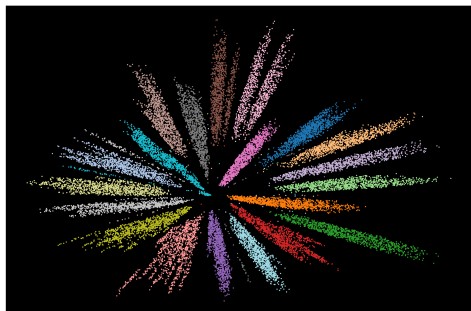

Figure 2: Two-dimensional representation of the embedding space generated by Hyper-Biome using UMAP. Each color corresponds to a different bacterial species.

## 5 CONCLUSIONS

This work introduces HyperBiome, a hyperbolic metric learning framework for encoding bacterial genomic databases. The method combines advanced tensorial sketching to efficiently represent $k$-mer distributions with the geometry of the Poincaré disk, thus capturing the intrinsic hierarchical structure of genomic data. A central contribution is the proposed multi-proxy loss, which leverages taxonomic organization to improve the reconstruction and representation of bacterial embeddings while generating learned proxies that summarize intra-species heterogeneity. As future work, we aim to extend proxy-based indexing to metagenomic applications, where species classification of sequencing data (commonly referred to as read binning) is pivotal and remains a significant challenge. More broadly, this study highlights hyperbolic metric learning as a promising paradigm for modeling bacterial diversity and domain-level organization, with the potential to accelerate large-scale exploration of the bacterial kingdom.

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
