# OpenReview forum: "The Bacterial Kingdom in Hyperbolic Space"
_ICLR.cc/2026/Conference — ICLR 2026 Conference Withdrawn Submission_

### Official Review · Reviewer_4XyE · 2025-10-27

**Soundness:** 1
**Presentation:** 1
**Contribution:** 1
**Rating:** 2
**Confidence:** 1

**Summary:**

The paper proposes HyperBiome, a metric learning framework that takes advantage of the geometry of the Poincaré ball to reconstruct the bacterial taxonomy.
Each genome is represented by a tensorial sketch of k-mer statistics.
A hierarchical multi-proxy loss organizes embeddings by both genus and species.
The experiments show that the proxy-based index achieves high accuracy, substantially reduces computational querying costs, and generalizes successfully to previously unseen species.

**Strengths:**

(i) Novelty: Leveraging hyperbolic space for taxonomic hierarchy.

(ii) The experimental results are positive, showing the effectiveness of their method.

(iii) Clear visualization: UMAP plots illustrate hierarchy preservation.

**Weaknesses:**

(i) The writing is very poor, especially since the presentation of the table is not beautiful.

(ii) No statistical uncertainty or robustness analysis in the experimental results.

**Questions:**

(i) Is there some statistical uncertainty or robustness analysis, e.g., the standard deviation?

(ii) The results in Table 1 do not show much difference,  e.g., 0.9969 vs. 0.9997. I am not sure whether such results can imply anything.

(iii) Can the writing be improved, especially the format of the table presentation?

(iv) The mathematical formulation is a little hard to understand. Could you please explain Section 2.3 to me?

Note: I do not fully understand the paper due to the writing. I keep myself open to improving my score if I fully understand the paper with the help of the authors or other reviewers.

**Details Of Ethics Concerns:**

I do not have ethical concerns.

---

### Official Review · Reviewer_6K3x · 2025-10-30

**Soundness:** 3
**Presentation:** 3
**Contribution:** 3
**Rating:** 6
**Confidence:** 2

**Summary:**

This article introduces HyperBiome, a hyperbolic metric learning framework designed to capture the hierarchical classification structure of bacterial genomes in a latent space. This method embeds genome representations into a Poincaré sphere, utilizing its tree-like geometry to encode relationships between species and genera more naturally than Euclidean embeddings. The method was validated on the AllTheBacteria dataset, demonstrating competitive accuracy, better scalability to large genome sets, and efficient classification.

**Strengths:**

- The paper is clearly written and well-structured, with solid motivation grounded in the growing scale and hierarchical organization of bacterial genomic databases.

- The proposed use of hyperbolic geometry to capture taxonomic hierarchy is conceptually sound and aligns well with the intrinsic tree-like nature of bacterial taxonomy, making the technical approach meaningful and biologically interpretable.

- The experimental setup is well-designed, leveraging a large-scale bacterial dataset (AllTheBacteria) and evaluating both representation quality and retrieval/classification efficiency. Furthermore, this paper demonstrates the good generalization ability of this method to unseen species, indicating its robustness and potential practical application value, which can be used in bacterial genome analysis workflows.

**Weaknesses:**

I am not an expert in this specific subfield, but I am slightly concerned about the benchmarking. The authors mention some related models that combine hyperbolic geometry and sequence data (e.g., NeuroSEED, HGE). Would the authors consider including these methods as baselines or comparisons to make the proposed model more convincing?

**Questions:**

Please see the weaknesses section for further discussion.

---

### Official Review · Reviewer_cCeE · 2025-10-31

**Soundness:** 2
**Presentation:** 1
**Contribution:** 1
**Rating:** 2
**Confidence:** 3

**Summary:**

HyperBiome introduces a metric-learning framework that exploits the non-Euclidean geometry of the Poincaré ball to model hierarchical relationships among bacterial genomes. The approach simultaneously learns embeddings at the species and genus levels and enables efficient, incremental updates without retraining the entire model.

The submission offers potentially valuable ideas; however, the work appears highly underdeveloped at this stage. Considerable refinement and additional experimentation is required to address the existing limitations and strengthen the overall contribution.

**Strengths:**

The paper extends early efforts demonstrating the promise of hyperbolic frameworks within genomic analysis. While the proposed approach offers an intriguing and potentially original direction for applying hyperbolic geometry to biological problems, a more comprehensive empirical and theoretical evaluation is needed to establish its validity and significance.

**Weaknesses:**

- The paper lacks a clear, high-level architectural overview. A schematic or block diagram illustrating the full pipeline would substantially improve clarity.

- Experiment 1 serves mainly as a hyperparameter search (over curvature c and clipping radius τ) rather than yielding substantive scientific insights. This analysis should be framed as an ablation or sensitivity study rather than a main experiment.

- The work does not explore or justify the choice of k-mer length. Because k critically affects representational granularity and computational cost, a systematic evaluation of different k values is needed to support the modeling decisions.

- The empirical analysis is relatively narrow in scope. Additional biological validation or interpretability analyses would strengthen the claim that the learned hyperbolic embeddings capture meaningful genomic organization beyond a single proxy accuracy metric.

- Experiment 3 is insufficiently defined. The experiment lacks a clear quantitative framework for evaluating the learned representation. Currently, the comparison relies on UMAP visualizations, which are sensitive to initialization, hyperparameters, and stochastic effects, making them an unreliable proxy for geometric evaluation. Instead, using something like Poincaré disk plots with geodesic overlays would provide more interpretable insights. Also, results should be averaged over multiple random seeds with confidence intervals to ensure robustness.

- There is a need for objective, geometry-aware metrics. Experiment 3 should incorporate quantitative measures that assess embedding quality directly in hyperbolic space, such as (1) class compactness and separation (e.g. within-class Fréchet variance), (2) label–embedding agreement (e.g. kNN accuracy), and/or (3) hierarchical fidelity (e.g. tree reconstruction accuracy).

Overall, the paper presents an intriguing and potentially valuable framework, but it currently lacks sufficient methodological clarity, architectural exposition, and experimental depth to support its main claims.

**Questions:**

- Why does the framework restrict the hierarchical modeling to only the species and genus levels? Would extending the hierarchical loss to higher taxonomic ranks (family, order, class) provide additional representational or biological benefits? It would be interesting to explore these tradeoffs further in the paper.

- The comparison to PanSpace (Experiment 2) appears limited to a single embedding dimensionality (for HyperBiome) and an otherwise fixed configuration. Given that this constitutes the main experimental result, could the authors provide a more comprehensive evaluation under varying hyperparameters and embedding sizes? Additionally, how does HyperBiome compare to PanSpace and other baselines in terms of training and inference efficiency? Including comparisons to simpler deep learning or metric-learning baselines could strengthen the empirical claims.

- How are new proxies generated and integrated into the model? Does proxy creation depend on the original set of trained proxies, or can it be performed independently using new embeddings?

---

### Official Review · Reviewer_FREF · 2025-11-02

**Soundness:** 3
**Presentation:** 3
**Contribution:** 3
**Rating:** 8
**Confidence:** 5

**Summary:**

The manuscript describes HyperBiome, a framewirk for for embedding of k-mer charcteristic verctors in a low dimensional hyperbolic space.
Representaiton includes genus- and species-level proxies
Hyperbiome is compared to a sOTA bacterial indexing method

**Strengths:**

* compelling idea of using hyperbolicity for an exponentially-diversifying evolutionary space
* demonsstrated non-euclidean curvature for the training (seen) data
* demonstrated generalaziability (performance on unseen data beyond SOTA
* demonstrated interpretability through UMAP visualization

**Weaknesses:**

1 there is no seprataion of the ideas potentially contributing
2 the seen data does not support hyperbolicity.
3 randomness ignored
4 no computational resources analysis
5. the number (2) of hierarchical levels is arbitrary
6. the data filtration leaves a very small dataset
7. No genus-level results

**Questions:**

1. There are multiple changes between the developed method (HyperBiome) and the comparison method (PanSpace):
        - kmer length
        - sketching (Hypergen - a cited, rather than new idea) vs. not
        - 2-level hierarchical proxies vs. not
        - hyperbolic space
   It is not clear which differnece(s) contribute to the performance gap. The authors are requested to experminetally prove the value of each change in order to claim any of them is important.

2. the ablation study for curvature c suggests the least-curved  (nearly Euclidean) space (c=0.01) is best for the seen data and panspace (c=0) is even better. This suggests something in the method and its objective funciton is off.  This often comes up when regularization facilitates test-set generalizability at the expense of train-set accuracy. However in this work regularization only comes up only in the context of clipping strategy and the less-regularized version prevails. The authors should find a consistent penalty funciton or otherwise explain what is going on here.

3. Sketching involves random subsetting, and so does the evaluation (by selecting the unseen set). The single-run results leave concern of cherrypicking. Repeating the analysis, say. 10 times with wach source of randomess, and reporting means + STDs would reassure this is not the case.

4. The motivation behind this field of investigation is that standard indexing methods (e.g. BLAST) are overly resource intensive. The manusript is self-defeating in not reporting anythign in terms of computational resources. Please compare to other methods (Hypergen, Panspace)

5. Please present results without hierarchy, and with additional phylogenetic layers (order, family)

6. The dataset that ends up being encoded is actually very small. Is this fliternign all that is needed to solve the resource problem? Please discuss

7. Please present accuracy at the genus level

8. The PanSpace UMAP in this manuscript differs markedly from what is reported in the PanSpace manuscript. Please explain.

---

### Note · Authors · 2025-11-13

**Comment:**

We would like to thank the reviewers for their valuable feedback and constructive comments on our submission. After careful consideration, we believe that our work can be significantly improved by incorporating their suggestions and performing further optimization and validation of HyperBiome. However, given the current time constraints and submission deadlines, we are unable to complete these improvements adequately. Therefore, we have decided to withdraw our paper at this stage and take the necessary time to refine our research.

**Withdrawal Confirmation:**

I have read and agree with the venue's withdrawal policy on behalf of myself and my co-authors.